# TCAD Analysis of Leakage Current and Breakdown Voltage in Small Pitch 3D Pixel Sensors

**DOI:** 10.3390/s23104732

**Published:** 2023-05-13

**Authors:** Jixing Ye, Abderrezak Boughedda, D M S Sultan, Gian-Franco Dalla Betta

**Affiliations:** 1Dipartimento di Ingegneria Industriale, Università degli Studi di Trento, 38123 Trento, Italy; jixing.ye@unitn.it (J.Y.);; 2Trento Institute for Fundamental Physics and Applications-Istituto Nazionale di Fisica Nucleare (TIFPA-INFN), 38123 Trento, Italy; 3Fondazione Bruno Kessler, 38123 Trento, Italy

**Keywords:** 3D pixel sensors, TCAD simulation, breakdown voltage

## Abstract

Small-pitch 3D pixel sensors have been developed to equip the innermost layers of the ATLAS and CMS tracker upgrades at the High Luminosity LHC. They feature 50 × 50 and 25 × 100 μm${}^{2}$ geometries and are fabricated on p-type Si–Si Direct Wafer Bonded substrates of 150 μm active thickness with a single-sided process. Due to the short inter-electrode distance, charge trapping effects are strongly mitigated, making these sensors extremely radiation hard. Results from beam test measurements of 3D pixel modules irradiated at large fluences ($10^{16}n_{eq}/{\mathrm{cm}}^{2}$) indeed demonstrated high efficiency at maximum bias voltages of the order of 150 V. However, the downscaled sensor structure also lends itself to high electric fields as the bias voltage is increased, meaning that premature electrical breakdown due to impact ionization is a concern. In this study, TCAD simulations incorporating advanced surface and bulk damage models are used to investigate the leakage current and breakdown behavior of these sensors. Simulations are compared with measured characteristics of 3D diodes irradiated with neutrons at fluences up to 1.5 × 10${}^{16}n_{eq}/{\mathrm{cm}}^{2}$. The dependence of the breakdown voltage on geometrical parameters (e.g., the n${}^{+}$ column radius and the gap between the n${}^{+}$ column tip and the highly doped p${}^{++}$ handle wafer) is also discussed for optimization purposes.

## 1. Introduction

The Large Hadron Collider (LHC) will experience an upgrade in the near future and step into the High Luminosity LHC (HL-LHC) era, in which the average number of proton–proton interactions per bunch crossing for the ATLAS experiment will increase to 140–200, and the integrated luminosity at the end of the ATLAS and CMS experiments is estimated to be 4000 $fb^{-1}$[1]. The double-sided 3D pixel detectors currently under operation in the ATLAS pixel Insertable B-Layer (IBL), which represent the radiation-hardest solution in practice at the moment [2], are not compatible with the HL-LHC scenario. In fact, their pixel size is 50×250
μm${}^{2}$, which would cause the occupancy to be too large. As a consequence, sensors with a smaller pixel size were deemed necessary in order to avoid event pile up; in 3D sensors, a smaller size also leads to smaller inter-electrode distance, thus providing a strong enough radiation hardness [3,4,5].

Small-pitch 3D pixel sensors, featuring pixel sizes of 50 × 50 and 25 × 100 μm${}^{2}$ with one readout electrode (hereafter referred to as 50 × 50 − 1*E* and 25 × 100 − 1*E*), were therefore designed to be compatible with the readout chips developed by the CERN RD53 Collaboration [6]. Fondazione Bruno Kessler (FBK), a pioneer foundry in the 3D sensor community for decades, successfully fabricated the first batch of small-pitch 3D pixel sensors based on single-sided process in 2015 [5]. Other R&D batches were later produced, followed by dedicated irradiation campaigns and beam tests, with results showing that a hit efficiency around 97% can be achieved after irradiation up to $1.8\times 10^{16}n_{eq}/{\mathrm{cm}}^{2}$ [7,8,9,10] at a maximum voltage of (∼150 V). These promising results paved the way for the use of 3D pixels to equip the innermost layers of the ATLAS and CMS Trackers at HL-LHC. However, the downscaled sensor structure lends itself to high electric fields as the bias voltage is increased, raising concerns regarding the onset of impact ionization effects, which could impact on the noise and the power dissipation, eventually leading to breakdown.

Nonetheless, it is possible to mitigate these problems by locating the origin of the breakdown and implementing layout solutions that are able to enhance the breakdown voltage. Therefore, Technology Computer-Aided Design (TCAD) simulations incorporating advanced surface and bulk damage models were performed based on the proposed geometries, and results were compared with measured characteristics of 3D diodes irradiated with neutrons at fluences up to 1.5 × 10${}^{16}n_{eq}/{\mathrm{cm}}^{2}$. Further TCAD simulations were carried out to study the influence that different geometrical parameters (e.g., the n${}^{+}$ column radius and the gap between the n${}^{+}$ column tip and the highly doped p${}^{++}$ handle wafer) have on the breakdown behavior of the sensors, providing useful information for optimizing the sensor layout.

The paper is organized as follows: Section 2 focuses on the description of the device under study, electrical tests and the simulation approach, and Section 3 covers the comparison between the experiments and simulations. The influence of different geometrical parameters on the breakdown behavior of the sensors is presented in Section 4, followed by conclusions in Section 5.

## 2. Experiment & Simulation Setup

### 2.1. Device Description

A cross-section of the devices is shown in Figure 1a. A 500-μm thick p${}^{++}$ handle wafer is directly bonded to a thin p${}^{-}$ sensor wafer to ensure a high mechanical stability [4]. Vertical electrodes are etched by Deep Reactive Ion Etching (DRIE) from the front side. The p${}^{+}$ electrodes penetrate into the handle wafer to allow for bias from the backside (the handle wafer is thinned as a post-processing step, and a metal layer is deposited to ease wire bonding). On the contrary, the etching of the n${}^{+}$ electrodes stops at a safety distance (gap = 30 μm) from the handle wafer to prevent early breakdown. A p-spray layer prevents the front surface from inversion, thus isolating the n${}^{+}$ electrodes. Thorough evaluations based on the signal amplitude, the sensor capacitance, and the characteristics of the readout chips have been carried out to optimize the thickness of the active volume, for which a value of 150 μm was finally chosen [9].

Figure 1b shows the layouts of the two pixel designs that will equip the ATLAS Inner Tracker (ITk), namely 50 × 50 − 1*E* and 25 × 100 − 1*E*. These two layouts have a relatively low density and are not critical from a lithographical point of view. However, while in the 50 × 50 − 1*E* pixel, the inter-electrode distance L∼ 35 μm is small enough to make it very radiation hard, in the 25 × 100 − 1*E* pixel, L∼ 51 μm was initially considered to be not small enough to ensure a high radiation tolerance. This motivated the parallel development of a more aggressive layout for the 25 × 100 pixel with two readout electrodes, the 25 × 100 − 2*E* [9]. In this latter structure, the unitary 3D cell is effectively 25 × 50 μm${}^{2}$, and the layout is very dense, which necessitated making the column diameter as small as possible. After several DRIE tests, a value of 5 μm was chosen at FBK and used so far in all R&D and pre-production batches. Although FBK demonstrated the feasibility of the 25 × 100 − 2*E* sensors [11], this geometry was not selected for the ATLAS ITk due to the manufacturing complexity and the consequent lower yield.

### 2.2. Electrical Tests

We have already reported on the electrical characterization of these devices both before irradiation and after irradiation (e.g., X-ray/γ-ray irradiation, neutron/proton irradiation) [5,9,12]. Here, we focus on the measurements for devices both before irradiation and after neutron irradiation. The neutron irradiation was performed at the TRIGA Mark II reactor at the Jozef Stefan Institute (Ljubljana, Slovenia) with fluences of $1\times 10^{16}n_{eq}/{\mathrm{cm}}^{2}$ and $1.5\times 10^{16}n_{eq}/{\mathrm{cm}}^{2}$. The uncertainty in the neutron fluences is within 10%. The γ-ray background generated during neutron irradiation causes only a limited Total Ionizing Dose (TID), and it is estimated to be around 0.1 Mrad per $10^{14}n_{eq}/{\mathrm{cm}}^{2}$ [13].

Since the ultimate purpose of this paper is the analysis of the leakage current and the breakdown voltage, we mainly focused on the I-V characteristics. As test structures, we used 3D diodes, which reproduce the electrode configurations and layout details (including the edge region) of their parent pixel sensors. As an example, Figure 2 shows a layout detail of a 3D diode of 50 × 50 − 1*E* geometry. The diode includes an array of pixels with all n${}^{+}$ electrodes shorted by a metal grid to a probe pad on the frontside, whereas the bias is applied from the backside. The device is terminated at the periphery by a slim edge, which consists of a multiple fence of ohmic columns extending from the active area toward the cut line. This structure limits the lateral extension of the depletion region spreading from the outermost pixels before it reaches the highly damaged cut region [14]. Due to their small size (∼2 mm${}^{2}$), 3D diodes are often free from process-related defects, making it possible to investigate the intrinsic properties of the different structures.

Non-irradiated samples were tested at the University of Trento using a probe station and a Keithley 4200A-SCS parameter analyzer. The measurements were performed at room temperature (24.5 ± 0.5 ${}^{\circ }$C). The reverse bias voltage was ramped from 0 to 200 V until breakdown with a step of 0.5 V. Neutron irradiated samples were measured at INFN Genova after mounting them on dedicated Printed Circuit Boards (PCBs). The tests were performed at −25 ${}^{\circ }$C inside a climate chamber [9]. Before being measured, devices were subjected to an annealing at room temperature for seven days.

### 2.3. TCAD Simulation Approach

In our previous studies, it has been shown that similar breakdown voltages are measured for small-pitch 3D pixel sensors before irradiation and after X-ray/γ-ray irradiation, i.e., in the presence of surface damage only [5,12]. These findings suggest that the n${}^{+}$ column tip is the cause of the breakdown, instead of the junction between the n${}^{+}$ column and the p-spray on the front-side surface. Therefore, it is important to locate the origin of the breakdown using TCAD simulations so as to face the problem accordingly. For this purpose, TCAD simulations based on 3D domains of the exact layouts were carried out using Synopsys Sentaurus [15]. Exploiting the inherent 3D cell symmetry, the simulated structures represent only one quarter of the pixels of the two layouts, in order to reduce the number of grid points (hence simulation time). The simulation domains for the two geometries are shown in Figure 3. The red region on the front side represents the n${}^{+}$ electrode, while the p${}^{+}$ electrode is on the opposite corner. The frontside surface layers (oxide, polySilicon, contact and metal) are not shown in the figures. It is worth mentioning that the back diffusion of boron from the p${}^{++}$ handle wafer (in blue at the bottom of the figures) is also considered, with a depth of 10 μm as estimated from SIMS and C-V measurements.

Simulations use typical models (e.g., effective intrinsic density, doping dependent Shockley–Read–Hall generation/recombination and mobility, high field saturation, etc.) and default values for most parameters but the minority carrier lifetimes, for which values of 1 ms were chosen, typical of FBK technology. Impact ionization effects are incorporated according to the avalanche model by Van Ovestraeten/De Man. To gain a comprehensive insight into the electrical behavior of the devices (mainly the breakdown voltage), we performed the simulations considering different situations: before irradiation, with surface damage only, and with the combined effect of surface and bulk damage. Radiation damage effects are described using the Perugia surface/bulk damage model [16,17] and the CERN bulk damage model [18]. In these models, radiation effects are incorporated by introducing deep-level traps, which are described in the Physics Section of the Sentaurus [15] command file with their energy, concentration (proportional to fluence by an introduction rate) and capture cross-sections for electrons and holes. As for the bulk damage, both Perugia and CERN models are tuned for p-type silicon and use two acceptor trap levels and one donor trap level, with different values of the relevant parameters. Details on the model implementation can be found in [19]. Simulations were performed at the temperatures for which the respective models were validated, i.e., −25 ${}^{\circ }$C for the Perugia model and −38 ${}^{\circ }$C for the CERN model.

## 3. Comparison between Measurements and Simulations

### 3.1. Pre-Irradiation

The comparison between experiments and simulations of the 50×50−1E and the 25×100−1E structures are shown in Figure 4a,b, respectively. The measurements were done on different devices of the same structures (Exp1, ..., Exp4), and the results were normalized to the same active volume used in the simulations (a quarter of a pixel). The breakdown voltage, denoted as $V_{bd}$ in the legends of the plots, was calculated based on the parameter
(1)$$k\left(I,V\right)=\frac{\Delta I}{\Delta V}\cdot \frac{V}{I}$$
looking at the minimum voltage value for which k exceeds a predefined threshold [20]. Since the measured I-V curves after neutron irradiation show a smooth rise of the current rather than an abrupt increase, a relatively low value of k = 2 was chosen to extract the breakdown voltage.

We can see that the simulations accurately predict the magnitude of leakage current and breakdown voltage, although the shape of the simulated curves does not perfectly match the experimental ones. This can be attributed to the simplified simulation domain, which adequately represents the core of the 3D diodes (array of pixels) but does not account for edge effects raising from the diode periphery. In fact, when the bias voltage is increased, the depletion region spreading from the peripheral pixels extends into the slim-edge region, so the leakage current increases. This is the main reason why the experimental I-V curves exhibit a non-negligible slope, whereas the simulated ones are almost flat before the onset of impact ionization effects. In addition, there is a non-negligible dispersion in the experimental curves, which can be ascribed to spatial fluctuations in several parameters such as bulk doping concentration, generation lifetimes, oxide charge density, interface state density, etc., and also to the presence of defects, in some devices. Simulations have been performed using nominal values for process-related parameters, aiming at the prediction of a general trend rather than fitting the individual curves. In this respect, the overall agreement between simulation and measurements can be considered good enough, indicating that TCAD simulations are reliable.

Comparing Figure 4a,b, it can be seen that the leakage currents are slightly lower and the breakdown voltages are slightly larger in 25 × 100 − 1*E* devices—a fact that can be attributed to the larger inter-electrode distance, leading to smaller electric fields. Besides the different breakdown voltages observed in the two designs, the origins of the breakdown before irradiation are also different. For 50 × 50 − 1*E*, the main contribution of the breakdown comes from the n${}^{+}$ column tip. This can be understood looking at Figure 5a, where the region with the highest impact ionization rate at voltage of 135 V (after the onset of the breakdown) is depicted; on the contrary, for 25 × 100 − 1*E*, the breakdown takes place at the junction between the n${}^{+}$ column and the p-spray on the front-side surface, as shown in Figure 5b, which was extracted at 146 V.

### 3.2. Surface Damage

As a matter of fact, previous studies have shown that the breakdown voltage remains roughly the same after the exposure to different levels of γ-ray irradiation for 50×50−1E [12]. This outcome is consistent with a breakdown occurring at the tip and not at the surface. In fact, the larger density of oxide charge mitigates the electric field peak at the junction between the n${}^{+}$ column and the p-spray. Similar results for 50×50−1E have been found in the simulations, as shown in Figure 6a. The temperature used for the simulations were kept unchanged, −25 ${}^{\circ }$C, in order to have a direct comparison between different situations. The breakdown voltage does not change significantly in the presence of surface damage, in good agreement with the experiments. Moreover, it remains the same for the two different doses used, which is consistent with a breakdown origin at the column tip.

Due to the fact that the breakdown shifts from the front-side surface to the tip in 25×100−1E after the introduction of surface damage, the breakdown voltage increases by ∼ 20 V, and it remains the same for both doses, as shown in Figure 6b. The shift of the simulated curve of the non-irradiated device toward a lower breakdown value observed in Figure 6b as compared to Figure 4b is due to the lower temperature.

### 3.3. Combined Effects

Though the main concern after neutron irradiation comes from Non Ionizing Energy Loss (NIEL) damage, it is still necessary to take into account the TID for the simulations in order to have a full picture to analyze the breakdown behavior of the device.

Among the most important effects of bulk damage is the introduction of defects with deep energy levels in the forbidden band gap, which behave as generation/recombination centers, leading to an increase of the leakage current proportional to the fluence [21].

Figure 7a shows the results of 50×50−1E after irradiation up to $1\times 10^{16}n_{eq}/{\mathrm{cm}}^{2}$, while the results of 25×100−1E are shown in Figure 7b. The measurements were done at T = −25 ${}^{\circ }$C, the same temperature used for the simulations with the Perugia model. The temperature adopted with the CERN model was −38 ${}^{\circ }$C. The leakage current was then normalized to the value that corresponds to −25 ${}^{\circ }$C using the Shockley-Read-Hall (SRH) model [22]
(2)$$I\left(I_{R}\right)=I\left(T\right)\cdot {\left(\frac{T_{R}}{T}\right)}^{2}exp\left[-\frac{E_{eff}}{2k_{B}}\left(\frac{1}{T_{R}}-\frac{1}{T}\right)\right]$$
where $E_{eff}$ is the effective band-gap energy of silicon, $T_{R}$ is the reference temperature, and $k_{B}$ is the Boltzmann constant.

We can see that the CERN model reproduces the leakage current with good accuracy when the bias voltage is low. However, when the bias voltage is increased, the model fails to predict the onset of impact ionization effects, thus underestimating the leakage current and overestimating the breakdown voltage. The Perugia model predicts the magnitude of the leakage current and of the breakdown voltage, but the shape of the I-V curve is different from the measured ones. In particular, a plateau in the I-V curve is soon reached due to the fact that the model underestimates the full depletion voltage, as was also found in a previous study [23]. Moreover, an abrupt increase in the current is observed at breakdown, whereas the measured curves exhibit a smooth rise.

Similar considerations apply to results for the other considered fluences. Figure 8a shows the results of 50×50−1E after irradiation up to $1.5\times 10^{16}n_{eq}/{\mathrm{cm}}^{2}$, while the results of 25×100−1E are shown in Figure 8b. In this case, the I-V curves obtained with the CERN model start to deviate from the measured ones at lower voltage: the agreement is good enough until ∼50 V, but then the shape of the simulated curves show a different concavity from the measured ones, overestimating the current in the intermediate voltage range, and finally underestimating it at large voltage. Impact ionization effects have an impact only at a very large voltage, leading to non-realistic breakdown voltage values. The Perugia model, on the other hand, yields results affected by the same type of inaccuracies observed at the lower fluence, with a shape of the I-V curves that differs significantly from the measured ones.

The reasons for the observed discrepancies between simulations and measurements are manifold, among which are the following:-There is a relatively wide dispersion in the experimental curves, especially for devices irradiated at $1\times 10^{16}n_{eq}/{\mathrm{cm}}^{2}$, possibly due to the uncertainties in the neutron fluence (within 10%) and in the measurement temperature (within 1 ${}^{\circ }$C, that corresponds to a 10% variation in the leakage current);-Edge effects also play a role, as observed in [9], although to a lesser extent with respect to the pre-irradiation case, because the extension of the depletion region into the slim-edge region is much smaller after irradiation;-Both the Perugia and CERN model have been developed with reference to the characteristics of planar sensors, and so far no attempt has been made to tune their parameters for using them with 3D sensors; moreover, as far as the I-V curves are concerned, both models have been validated up to fluences smaller than those involved in our study, i.e., $3\times 10^{15}n_{eq}/{\mathrm{cm}}^{2}$ for the Perugia model and $8\times 10^{15}n_{eq}/{\mathrm{cm}}^{2}$ for the CERN model;-As far as the specific deficiencies in the two models, the Perugia model largely underestimates the depletion voltage, so the I-V curves rapidly saturate, leading to an overestimation of the leakage current, whereas the CERN model provides a better agreement with the shapes of the experimental I-V curves, but then does not properly describe the effects related to impact ionization, which are predicted to occur at a much larger voltage than that observed experimentally.

A summary of the simulated and measured breakdown voltages is reported in Table 1. Despite the previously mentioned problems, the breakdown voltages predicted by the Perugia model are in closer agreement with the experimental ones, especially for the 25×100−1E geometry.

The geometric current damage constant $\alpha^{★}$ [24] was calculated based on
(3)$$\alpha^{★}=\frac{I}{V\cdot \Phi }$$
where *I* corresponds to the current measured at a reverse bias of 100 V (which is supposed to ensure full depletion) and scaled to a reference temperature of 20 ${}^{\circ }$C using Equation (Equation 2), *V* is the geometric volume, and Φ is the fluence. The simulated and measured current damage rates are listed in Table 2.

The reported values of course reflect the differences observed in the I-V characteristics, as commented upon previously. The CERN model provides a better agreement with the experimental values, as well as with the generally accepted reference value, that is $∼4.64\times 10^{-17}$ A/cm considering an annealing of seven days at room temperature [25]. To be specific, the agreement is particularly good for the 25×100−1E geometry (within 6%), whereas a larger discrepancy (within 17%) is found for the 50×50−1E geometry at $1.5\times 10^{16}n_{eq}/{\mathrm{cm}}^{2}$. In the case of the 50×50−1E geometry at $1.0\times 10^{16}n_{eq}/{\mathrm{cm}}^{2}$, the discrepancy is much larger (within 29%), but it should be stressed that at the considered voltage of 100 V, impact ionization effects are evident in the experimental curve, and thus the reported $\alpha^{★}$ is not representative of a pure thermally generated current.

Apart from the case of 50×50−1E geometry at $1.0\times 10^{16}n_{eq}/{\mathrm{cm}}^{2}$, due to the above-mentioned reason, the Perugia model overestimates the $\alpha^{★}$ values by up to 43% in the worst case, that is, the 25×100−1E geometry at $1.5\times 10^{16}n_{eq}/{\mathrm{cm}}^{2}$, but it should be noted that at this high fluence, the considered voltage of 100 V might be not sufficient to reach full depletion in this geometry, leading to an underestimation of the experimental value.

## 4. Influence of Geometrical Parameters

Having observed from the simulations that the tip tends to limit the breakdown voltage in the presence of mixed surface/bulk damage, the study was extended to consider the impact of different geometrical parameters on the breakdown behavior of different designs. A systematic simulation of the two layouts was performed using different values of the gap and the electrode radius. For the gap, the effective value (i.e., the geometrical gap minus the boron back diffusion depth) was varied in the range from 15 μm to 30 μm. The latter value corresponds to the minimum column depth allowed by the ATLAS ITk specification. For the radius, it was varied in the range from 2 μm to 4 μm. Again, the latter value corresponds to the maximum radius allowed by the ATLAS ITk specification. Similarly to the previous section, the simulation is divided into three conditions: pre-irradiation, surface damage and bulk damage.

### 4.1. Pre-Irradiation

Figure 9a shows the simulation results of 50×50−1E before irradiation. We can see that the breakdown voltage increases as the gap and radius increase. The increase with the gap is more pronounced at small values; then, saturation is reached at larger gaps, since the surface becomes more and more critical. The radius impacts on the breakdown voltage both when it is originated at the tip and at the surface, but its effect also shows a saturation trend.

A similar trend is observed in 25×100−1E, as can be seen on Figure 9b, with sightly larger breakdown values. Here, the saturation of the gap effect is anticipated to 20 μm (i.e., the same value applying to the existing devices), which is reasonable, considering that the origin of the breakdown is on the front-side surface at that point.

### 4.2. Surface Damage

Although 3D detectors are normally not used in applications involving only surface damage, it is still interesting to observe the breakdown behavior in this condition. In fact, it makes it possible to study the dependence of the breakdown originating at the column tip on the geometrical parameters without the effects of the bulk damage. Figure 10a shows the simulation results of 50×50−1E after surface damage. We can see a clear dependence of the breakdown voltage on the radius and on the gap. A larger radius/gap corresponds to a higher breakdown voltage. Moreover, both doses used in the simulations gave the same breakdown voltage, which further proves that the breakdown is at the tip after surface damage.

The same conclusion can be drawn for 25×100−1E, as can be seen from Figure 10b. Moreover, because of larger inter-electrode distance, 25×100−1E has higher breakdown voltages than 50×50−1E.

### 4.3. Bulk Damage

TCAD simulation can be very time consuming for 3D structures, and it scales with the number of mesh points. Most importantly, it is necessary to use a very dense mesh on the front-side surface in order to have a precise description of it; this contributes more than one third of the total mesh points. Therefore, it is worthwhile to explore if the surface would cause a big difference on the breakdown voltage after radiation damage. Figure 11 shows the results for 25×100−1E, with reference to the nominal values of radius (2.5 μm) and gap (20 μm). Simulations were performed using the Perugia bulk damage model, which proved to yield a more accurate estimate of the breakdown voltage. The temperature was set to −25 ${}^{\circ }$C, and the TID was adjusted according to the fluence (10 Mrad when the fluence is $1\times 10^{16}n_{eq}/{\mathrm{cm}}^{2}$, and 20 Mrad when the fluence is $2\times 10^{16}n_{eq}/{\mathrm{cm}}^{2}$). Other key parameters are shown in the legend.

It can be seen that the oxide does not have any influence on the breakdown voltage under all simulated conditions and indeed has a very limited influence on the leakage current. It was also checked that the same trend applies to different geometrical parameters (e.g., radius = 2 μm) and layouts, namely 50×50−1E. Simulations were also repeated using the CERN bulk damage model, leading to the same conclusion. Therefore, it was deemed appropriate to remove the surface layers (p-spray, oxide) to study the breakdown prediction after radiation damage, as a way to boost the whole simulation process. Hence, we re-built the simulation structures without the surface and ran the simulation using different geometric parameters and different fluences.

Figure 12a shows the results for 50×50−1E when the fluence is $1\times 10^{16}n_{eq}/{\mathrm{cm}}^{2}$, while Figure 12b shows the results when the fluence is $2\times 10^{16}n_{eq}/{\mathrm{cm}}^{2}$. It can be seen that the breakdown voltage increases with the radius and the gap; higher fluence leads to higher breakdown voltage. As an example, it is possible to improve the breakdown voltage of the current design (gap = 20 μm) by more than 10 V, by increasing the column radius by one micron, from 2.5 μm to 3.5 μm. Indeed, this would slightly increase the capacitance and the dead volume, but it would still be within the ITk specification.

Figure 13a shows the results for 25×100−1E when the fluence is $1\times 10^{16}n_{eq}/{\mathrm{cm}}^{2}$, while Figure 13b shows the results when the fluence is $2\times 10^{16}n_{eq}/{\mathrm{cm}}^{2}$. The behavior is very similar to the one observed for the 50×50−1E: a larger radius and gap corresponds to a larger breakdown voltage. As an example, keeping the gap at 20 μm, the breakdown voltage increases by 20 V after raising the column radius by 1 μm from 2.5 μm to 3.5 μm.

## 5. Conclusions

Simulations based on the exact geometries of 50×50−1E and 25×100−1E 3D pixels were performed and compared with the measured characteristics. The inspection of ionization maps proved that the breakdown occurs at the n${}^{+}$ column tip for 50×50−1E and at the surface for 25×100−1E before irradiation; it remains at/shifts to the tip in the presence of surface damage. Moreover, the 25×100−1E has a higher breakdown voltage due to larger inter-electrode distance. Before irradiation, the agreement between simulated and measured characteristics is quite good, with the observed discrepancies being mainly due to edge effects, not considered in the simulations, and to the relatively wide dispersion of the experimental curves. After irradiation, the CERN model provides a relatively good agreement with measurements in terms of leakage current, but largely overestimates the breakdown voltage. On the contrary, the Perugia model is less accurate in predicting the leakage current but provides a reasonably good agreement with the measured breakdown voltages (within 28%). Nevertheless, the shape of the I-V curves in the presence of bulk damage is different from what has been measured, meaning that the models have to be further improved to provide a higher accuracy, both in terms of leakage current and breakdown behavior.

Furthermore, a systematic study of the impact of different geometrical parameters on the breakdown voltage has been conducted. After irradiation, the origin of the breakdown is always at the tip regardless of the layout and geometrical configurations. It would be possible to improve the breakdown voltage by increasing the gap between the column tip and the handle wafer, but the effect soon saturates at gap values of practical interest. On the contrary, an improvement in the breakdown voltage of ∼ 10 V and ∼ 20 V can be obtained for the 50×50−1E and 25×100−1E, respectively, by increasing the electrode radius by 1 μm, while keeping the effective gap at 20 μm. This would be particularly important to extend the operational margin in the 25×100−1E, which, due to its larger inter-electrode distance, requires higher bias voltage to be full efficient. Given the relatively low density of the pixel layout, this solution is feasible with a minor impact on the sensor capacitance and geometrical efficiency, and it will be implemented in the next batch of 3D pixels at FBK.

## Figures and Tables

**Figure 1 sensors-23-04732-f001:**
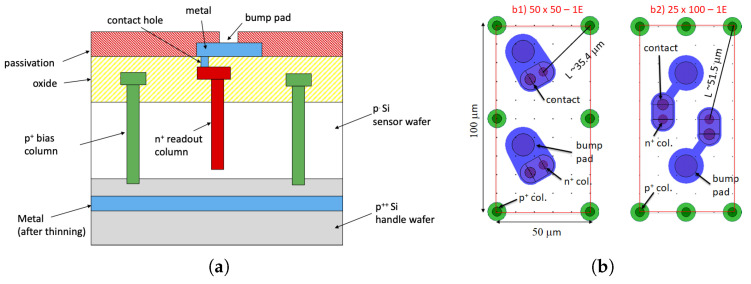
Description of the devices under study: (**a**) schematic cross-section, and (**b**) layouts of different designs (two adjacent pixels are shown).

**Figure 2 sensors-23-04732-f002:**
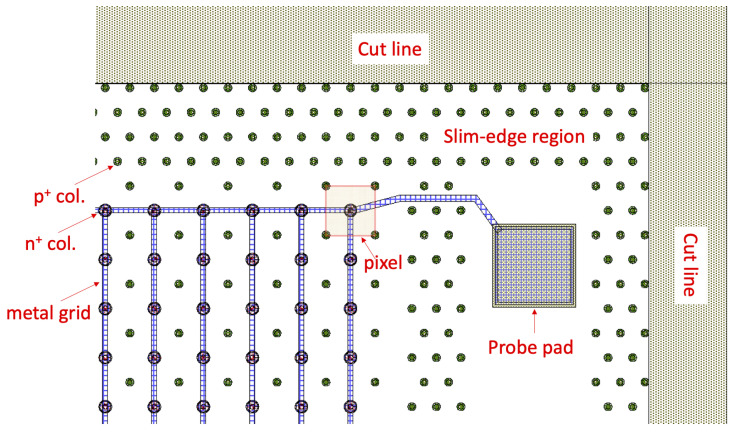
Layout of a 3D diode of 50 × 50 − 1*E* geometry: detail of a corner.

**Figure 3 sensors-23-04732-f003:**
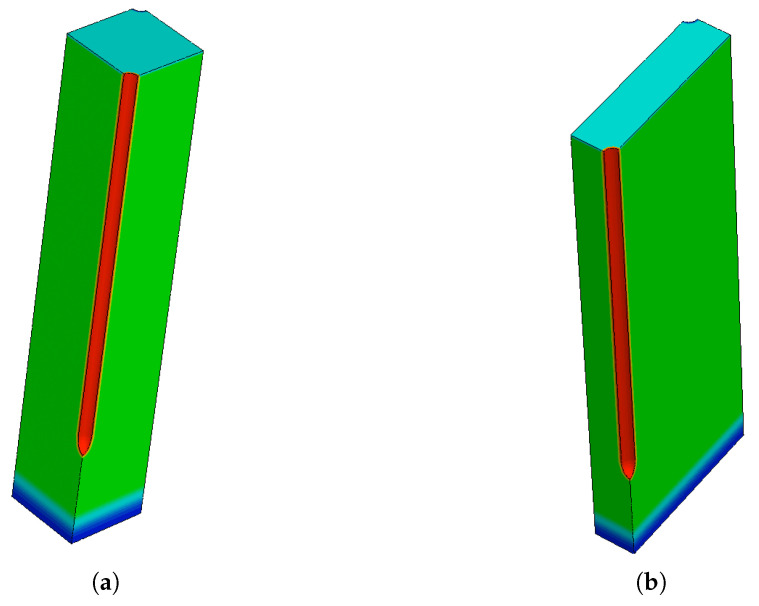
Simulation domain of different layouts: (**a**) 50 × 50 − 1*E*, and (**b**) 25 × 100 − 1*E*.

**Figure 4 sensors-23-04732-f004:**
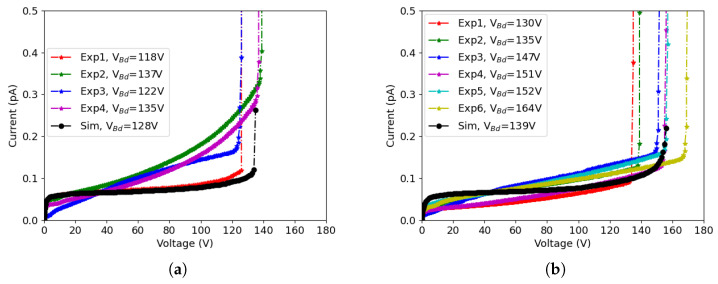
μmated (Sim) I-V curves before irradiation at room temperature, with breakdown voltages $V_{bd}$ reported in the legend: (**a**) 50 × 50 − 1*E*, and (**b**) 25 × 100 − 1*E*.

**Figure 5 sensors-23-04732-f005:**
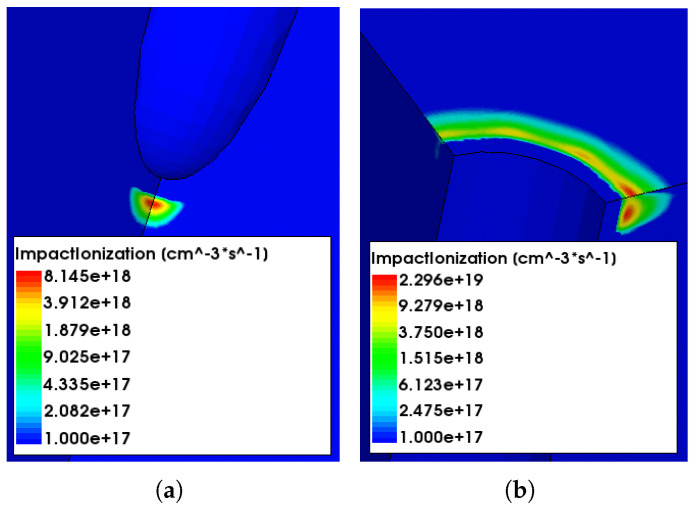
Impact Ionization map of different layouts before irradiation: (**a**) 50 × 50 − 1*E* at 135 V, and (**b**) 25 × 100 − 1*E* at 146 V. Results point to a different origin of the breakdown in the two structures before irradiation.

**Figure 6 sensors-23-04732-f006:**
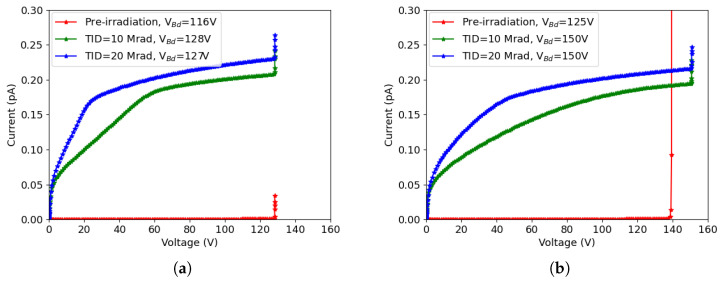
Simulated I-V curves of different layouts both before and after surface damage at −25 ${}^{\circ }$C: (**a**) 50×50−1E, and (**b**) 25×100−1E.

**Figure 7 sensors-23-04732-f007:**
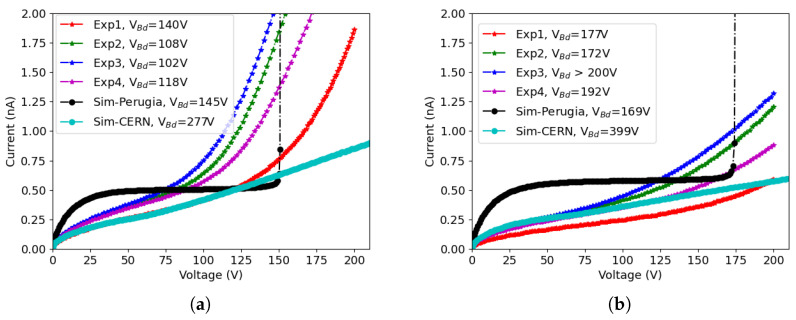
Comparison between measured (Exp) and simulated (Sim) I-V curves of different layouts after irradiation at a fluence of $1\times 10^{16}n_{eq}/{\mathrm{cm}}^{2}$, with breakdown voltages $V_{bd}$ reported in the legend: (**a**) 50×50−1E, and (**b**) 25×100−1E.

**Figure 8 sensors-23-04732-f008:**
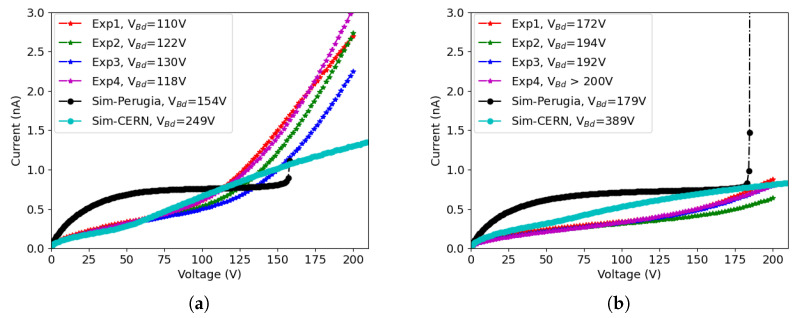
Comparison between measured (Exp) and simulated (Sim) I-V curves of different layouts after irradiation at a fluence of $1.5\times 10^{16}n_{eq}/{\mathrm{cm}}^{2}$, with breakdown voltages $V_{bd}$ reported in the legend: (**a**) 50×50−1E, and (**b**) 25×100−1E.

**Figure 9 sensors-23-04732-f009:**
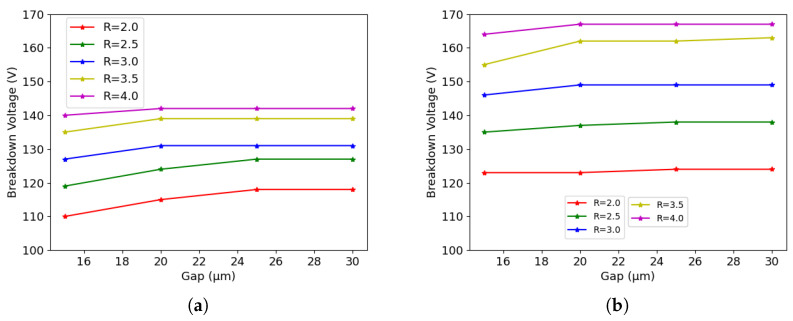
Breakdown voltage acquired using different geometrical parameters before irradiation: (**a**) 50×50−1E, and (**b**) 25×100−1E.

**Figure 10 sensors-23-04732-f010:**
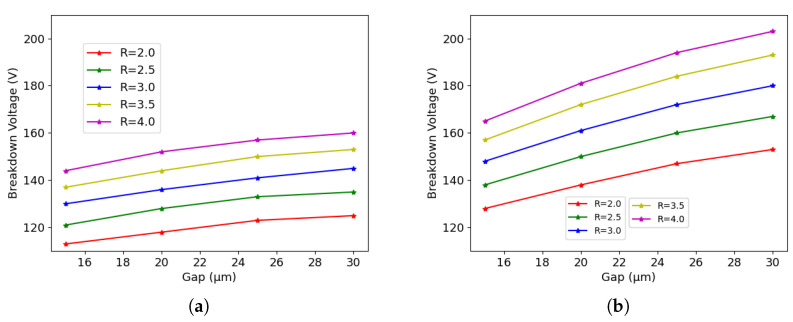
Breakdown voltage acquired with TID = 10/20 Mard: (**a**) 50×50−1E, and (**b**) 25×100−1E.

**Figure 11 sensors-23-04732-f011:**
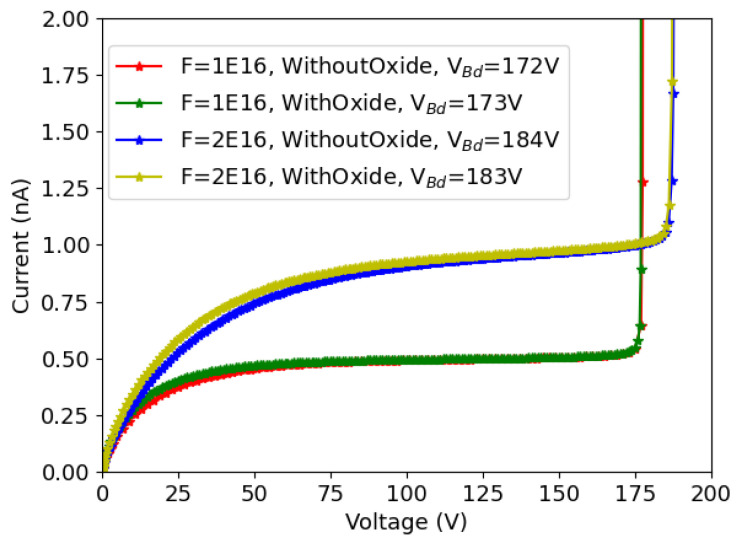
The effect of the oxide and p-spray have on the breakdown voltage, 25×100−1E.

**Figure 12 sensors-23-04732-f012:**
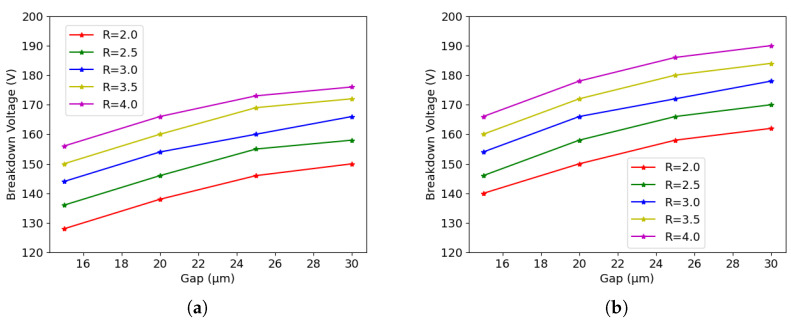
Influence of different geometrical parameters for 50×50−1E after bulk damage: (**a**) $1\times 10^{16}n_{eq}/{\mathrm{cm}}^{2}$, and (**b**) $2\times 10^{16}n_{eq}/{\mathrm{cm}}^{2}$.

**Figure 13 sensors-23-04732-f013:**
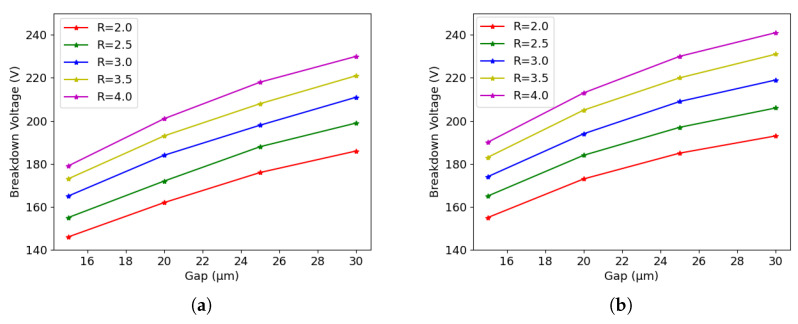
Influence of different geometrical parameters for 25×100−1E after bulk damage: (**a**) $1\times 10^{16}n_{eq}/{\mathrm{cm}}^{2}$, and (**b**) $2\times 10^{16}n_{eq}/{\mathrm{cm}}^{2}$.

**Table 1 sensors-23-04732-t001:** Summary of the breakdown voltages after neutron irradiations. For the experiment, the average ± standard deviation values are reported.

Fluence $\left(10^{16}n_{\mathrm{eq}}/{\mathrm{cm}}^{2}\right)$	Structure	$V_{\mathrm{bd}}$ (V) Experiment	$V_{\mathrm{bd}}$ (V) Perugia Model	$V_{\mathrm{bd}}$ (V) CERN Model
1.0	50×50−1E	117 ± 14.5	145	277
	25×100−1E	180 ± 8.5	169	399
1.5	50×50−1E	120 ± 7.2	154	249
	25×100−1E	186 ± 9.9	179	389

**Table 2 sensors-23-04732-t002:** Summary of current damage constant after neutron irradiation. For the experiment, the average ± standard deviation values are reported.

Fluence $\left(10^{16}n_{\mathrm{eq}}/{\mathrm{cm}}^{2}\right)$	Structure	$\alpha^{*}$ Experiment $\left(10^{-17}A/\mathrm{cm}\right)$	$\alpha^{*}$ Perugia Model $\left(10^{-17}A/\mathrm{cm}\right)$	$\alpha^{*}$ CERN Model $\left(10^{-17}A/\mathrm{cm}\right)$
1.0	50×50−1E	6.92 ± 1.14	5.92	4.90
	25×100−1E	4.25 ± 0.91	5.74	4.22
1.5	50×50−1E	4.41 ± 0.36	5.91	5.14
	25×100−1E	3.87 ± 0.43	5.54	4.09

## Data Availability

Data are available upon request.

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
