# Peer review of "TCAD Analysis of Leakage Current and Breakdown Voltage in Small Pitch 3D Pixel Sensors"

_sensors, 2023, doi:10.3390/s23104732_

Round 1

Reviewer 1 Report

This article describes the results of TCAD simulations of 3D silicon sensors, which are the proposed technology for the innermost region of the Atlas tracker, and could have similar use-cases for other particle physics experiments, especially in future. Various comparisons are made: between simulation and laboratory data (for different prototype devices), between different geometries in simulation, between unirradiated and irradiated devices, and between different models for radiation damage. This will provide useful input for the field of silicon detector development, especially where radiation hardness is a priority. 

The context and methods are described well. Results are clearly presented and described, and the paper is well written.

Overall there is mixed agreement in the main features observed in simulation and real data. I don't think this justifies the statement in the abstract that the agreement is 'satisfactory' - there are clear discrepancies in particular after irradiation as shown in figures 6-7. I would recommend that this and other statements about data/TCAD agreement are qualified appropriately. One clear message from the paper is that further work on radiation damage models will be essential in order to trust the results of simulation with high confidence.

Similarly, on L201 (end of sec 3) the agreement between data and simulation for the geometric current damage constant is stated to be 'good enough' without qualification. I suggest that this is made more quantitative by reference to the quoted experimental uncertainties.

I also note that when making data/sim comparisons, there is not a common bias voltage for the two categories. For example, Fig 3 shows IV curves for data at bias voltages of 118,122,135,137V, while the simulation has a bias voltage of 128V. This is unfortunate but doesn't materially affect the conclusions. I suggest that the authors aim to be consistent for future comparisons.

In general the writing is good. I have a few small suggestions:

L18-19 - confusing repetition of 'collision / collisions'. I suggest editing e.g. "in which the average number of proton-proton interactions per bunch crossing will increase to 200"

L20 - "by the end of the experiment" is vague - you are here referring to the HL-LHC which includes multiple experiments. I suggest that you specify Atlas and CMS experiments.

L28: 'refer' -> 'referred'

Author Response

We would like to thank the reviewer for the insightful comments that allowed to improve the paper quality. We have carefully considered all comments and requests, and modified the paper accordingly, as detailed in the attached responses. The changes are also highlighted within the manuscript.

Reviewer 2 Report

The author proposed the TCAD analysis of leakage current and breakdown voltage in small pitch 3D pixel sensors based on a microstructured pn junction. Radiation damage effects are described using the Perugia surface/bulk damage model and the CERN bulk damage model. Unfortunately, referring to results in figure 6 and 7 comparing experimental results and simulation after irradiation there is not a good agreement between them. The authors should carefully explain why there is this strong difference.

Here other comments:

p.3. r.128-129: Not clear how the models are implemented to evaluate the effect of the ion damage on the pixel. Could the author explain how the model are implemented? Could the authors report in the experimental section some details about the simulations? See supplementary material of (https://doi.org/10.1002/aelm.202000615.

Figure 3: Why is the leakage current measured high than simulated? This is an observation also observed in other papers with TCAD-simulated pnjunctions (https://doi.org/10.1002/aelm.202000615). When the authors refer to exp1,...,4 are they referring to different devices or different measurements on the same device?

Figure 5: Why the simulated leakage current is higher after irradiation then before irradiation?

Figure 6-7: How the authors have extrapolated the measured breakdown voltages? There is not a clear variation in the slope of the IV curve. Simulation perfomed with CERN and Perugia models differ from experimental results. Could the authors explain why these models are not able to match the experimental results before irradiation (figure 3) but not after irradiation (figure 6-7)?

It's required a revision of the manuscript.

Author Response

(The authors gave the same response as above.)

Round 2

Reviewer 2 Report

The authors proposed a manuscript with the title "TCAD analysis of leakage current and breakdown voltage in small pitch 3D pixel sensors".Also after a major revision, in opinion of this reviewer the discrepancies between experimental results and simulations seem to indicate that the model used (CERN and Perugia) are not correctly describing the investigated phenomena (see figures 7 and 8).